

# The optimization and operation of multi-energy-coupled microgrids by the improved fireworks algorithm-shuffled frog-leaping algorithm

Xubo Yue[1], Jing Zhang[2], Junhui Guo[2], Jianfei Li[2] and Diyu Chen[2]

[1] Taizhou Hongchuang Group, Taizhou, China
[2] Taizhou Hongyuan Electric Power Design Institute, Taizhou, China

## ABSTRACT

This study aims to address optimization and operational challenges in multi-energy coupled microgrids to enhance system stability and reliability. After analyzing the requirements of such systems within comprehensive energy systems, an improved fireworks algorithm (IFWA) is proposed. This algorithm combines an adaptive resource allocation strategy with a community genetic strategy, automatically adjusting explosion range and spark quantity based on individual optimization status to meet actual needs. Additionally, a multi-objective optimization model considering active power network losses and static voltage is constructed, utilizing the shuffled frog-leaping algorithm (SFLA) to solve constrained multi-objective optimization problems. Through simulation experiments on a typical northern comprehensive energy system, conducted with a scheduling period of T = 24, the feasibility and superiority of IFWA-SFLA are validated. Results indicate that IFWA-SFLA performs well in optimizing microgrid stability, managing electrical energy flow effectively within the microgrid, and reducing voltage fluctuations. Furthermore, the circuit structure and control strategy of microgrid energy storage bidirectional inverters based on IFWA are discussed, along with relevant simulation results.

Corresponding author
Jianfei Li, Ljfei23@126.com

## INTRODUCTION

In contemporary energy systems, microgrids serve a vital role as small-scale, sustainable energy systems, catering to local energy demands and improving energy utilization efficiency. Comprising diverse energy resources like solar and wind energy, alongside batteries, microgrids integrate with various loads, including residential and commercial buildings. Through intelligent control and optimized operation, microgrids can achieve coordinated operation and configuration among multiple energy sources, enhancing system stability and reliability. However, optimizing and operating multi-energy-coupled microgrid systems present several challenges, including effectively managing and scheduling diverse energy sources while reducing system losses and maintaining stable voltage levels. Therefore, research into optimizing and operating these systems holds

significant theoretical and practical importance (*Yang et al., 2020*; *Arfeen et al., 2021*; *Sheng et al., 2023*). Microgrids are designed to provide electricity, heat, and cooling to fulfill diversified energy needs, particularly in decentralized or remote areas and urban regions. However, effectively operating and optimizing microgrid operation present complex challenges, especially when considering multiple energy sources, various electrical loads, and ever-changing environmental conditions. In the current energy landscape, the widespread adoption of renewable energy resources has become a significant trend (*Li et al., 2020*; *Liu, Khan & Yuan, 2023*; *Shen et al., 2021*). The rapid development of renewable energy sources such as solar and wind power makes microgrids an ideal choice for achieving clean, low-carbon, and efficient energy supply (*Wang et al., 2023*). Nevertheless, renewable energy sources exhibit intermittency and uncertainty, posing challenges to the stability and reliability of microgrids. In order to overcome these issues, multi-energy-coupled microgrids have been introduced, integrating various energy resources (*e.g.*, solar, wind, biomass) and energy conversion technologies (*e.g.*, photovoltaics, wind turbines, fuel cells) simultaneously to enhance energy supply stability and reliability.

The optimization and operation of multi-energy-coupled microgrids entail multiple objectives, including reducing energy costs, decreasing carbon emissions, enhancing system reliability, and maximizing energy utilization. Efficient optimization algorithms are essential to achieve these goals. The fireworks algorithm (FWA) and frog leaping algorithm (FLA) are two nature-inspired optimization methods that have demonstrated promise in various fields (*Rezaee Jordehi, 2020*; *Venkatesan et al., 2021*; *Lotfi, 2022*). However, in the complex multi-objective optimization problems associated with microgrid operation, traditional FWA and FLA encounter challenges such as limited local search capabilities, convergence speed, and global search performance. During the integration of microgrids into the grid, various distributed generation methods must be considered, such as gas turbines and fuel cells, accounting for their energy efficiency, generation costs, and actual capacity. Additionally, the cost and environmental impact of these distributed energy sources should be weighed to make informed and cost-effective decisions. The microgrid's network structure also requires reconfiguration to ensure the efficient utilization of nearby energy resources (*Xue & Wu, 2019*; *Macedo et al., 2021*). Furthermore, potential impacts on load power security and stability following the connection of distributed energy sources should be minimized through proper configuration.

*Velasco, Guerrero & Hospitaler (2024)* highlighted the exponential growth in the number of new algorithms in recent years and raised questions about some of them. The authors analyzed 111 recent studies and found that only 43% of the articles mentioned the "no free lunch" theorem. Additionally, they discovered that 65% of the research proposed improved versions of existing algorithms. This study emphasized the ongoing issue of the increasing number of existing algorithms and called for a more cautious approach to the introduction of new algorithms, suggesting innovation based on existing algorithms. *Aranha et al. (2022)* identified some problems with newly proposed metaphor-based metaheuristic algorithms. The authors criticized some algorithms for lacking substantive innovation and merely proposing new metaphorical concepts for the sake of publication.

Moreover, they pointed out issues with experimental validation and comparison, highlighting biases in experimental design and comparison in some articles. *Tzanetos & Dounias (2021)* presented some suggestions for natural-inspired optimization algorithms and discussed some current issues. The authors argued that some newly proposed natural-inspired algorithms fail to fully draw from successful strategies in natural systems and lack a practical application foundation. The article called for a more rational and cautious approach to the introduction of new algorithms and proposed some guiding principles for the development of natural-inspired algorithms. In summary, these three articles critically analyzed the introduction of new algorithms, pointing out problems and challenges, including a lack of substantive innovation, insufficient experimental validation, and a weak foundation for algorithm application. Therefore, the introduction of new algorithms requires more careful and rational consideration, with innovation based on existing algorithms, and scientific rigor in experimental validation and comparison.

Multi-energy-coupled microgrids are pivotal for integrating diverse energy resources, enhancing system stability and reliability, thereby ensuring sustainable, dependable, and efficient energy supply. Microgrid operation entails diverse objectives, including cost reduction, emissions mitigation, and reliability enhancement, necessitating efficient optimization algorithms to achieve viable solutions. This study aims to enhance the fireworks algorithm and amalgamate it with the shuffled frog leaping algorithm (SFLA) to tackle the optimization and operational hurdles of multi-energy-coupled microgrids. Initially, an improved fireworks algorithm (IFWA) is proposed, integrating adaptive resource allocation strategies with community inheritance strategies. This entails automatic adjustment of the explosion range and spark quantity based on the optimization status of individuals within their communities, tailored to individual real-world requirements. Subsequently, a multi-objective optimization model is formulated, considering active power network loss and static voltage, and solved using SFLA for multi-objective optimization problems with constraints. Finally, through instance simulations, the efficacy of the proposed IFWA-SFLA algorithm in optimizing dispatch for comprehensive energy system microgrids is validated.

This study introduces several novel aspects: Firstly, it proposes a novel multi-objective optimization method, combining IFWA with SFLA for the first time. This fusion integrates the strengths of both algorithms, enhancing global search capability and local search accuracy through adaptive resource allocation and community genetic strategies, effectively addressing optimization problems in multi-energy-coupled microgrids. Secondly, considering the characteristics of multi-energy-coupled microgrids, the article constructs a multi-objective optimization model that simultaneously considers active power network losses and static voltage, a less common approach in previous studies. This model enables a more comprehensive evaluation and optimization of microgrid operational status, thereby improving system stability and reliability. Thirdly, through a case study in a typical comprehensive energy system in northern China, the article verifies the feasibility and superiority of the proposed algorithm, providing valuable references for practical application. Fourthly, the article tests the algorithm's performance not only in practical microgrid systems but also on classic nonlinear functions (such as Sphere and

Schaffer functions), comprehensively evaluating its ability to solve complex optimization problems, demonstrating versatility and efficiency. Finally, the article comprehensively evaluates the algorithm's performance, including key indicators such as optimal solution value, convergence speed, and computation time. This evaluation method provides a quantitative basis for analyzing the algorithm's performance, facilitating a deeper understanding of its strengths and areas for potential improvement. In summary, this study offers significant novelty in algorithm design, optimization model construction, practical application case verification, and performance evaluation, providing new solutions and theoretical support for optimizing and operating multi-energy-coupled microgrids.

## RELATED WORK

In the field of microgrid optimization and operation, many scholars have conducted related research and achieved some significant results. Particularly in the realm of multi-energy-coupled microgrids, some studies have focused on addressing issues such as energy scheduling, energy management, and system optimization in microgrid systems. In previous research, *Guerraiche et al. (2023)* employed metaheuristic algorithms to solve optimization problems in microgrid systems, proposing an ideal method for energy management in hybrid microgrid systems to optimize the series-parallel energy system. *Mahmoud et al. (2023)* concentrated on the optimization problem of microgrid energy management systems, minimizing the reduction of loads in each microgrid and further reducing the energy shortage in multi-microgrid systems through energy exchange between individual microgrids. The remaining shortage can be further reduced with the help of resources from community microgrids. The proposed optimization model is formulated as a nonlinear complex optimization problem, which can be more effectively handled through metaheuristic techniques compared to traditional analytical methods. In this article, a political optimizer is used. Additionally, *Zaki et al. (2023)* combined hybrid driving training-based and particle swarm optimization (HDTPSO) techniques to improve the transient voltage response when facing disturbances, enhancing its performance. Finally, two optimization techniques (HDTPSO and PSO) were compared, and the results showed that using the HDTPSO model for controller optimization produced a better impact. In addition to metaheuristic algorithms, some studies have employed model-based optimization methods to address optimization problems in microgrid systems. For example, *Mannini et al. (2024)* used model predictive control methods for energy management and optimization scheduling in microgrids. These studies establish dynamic system models and predict future load demand and energy supply conditions to formulate optimal control strategies, thereby achieving stable operation and maximizing energy utilization in microgrid systems. In summary, previous research has made some important achievements in microgrid optimization and operation, but there are still some challenges and issues to be addressed. The contribution of this study lies in introducing improved fireworks and frog-leaping algorithms and combining them with the characteristics of multi-energy-coupled microgrid systems to address some key issues in system

optimization and operation, thereby enhancing the stability and reliability of microgrid systems.

# COMPREHENSIVE ENERGY SYSTEM MULTI-ENERGY-COUPLED MICROGRID OPTIMIZATION

## Comprehensive energy system architecture

The rapid growth of renewable energy underscores the significance of comprehensive energy systems, integrating diverse energy resources and technologies for sustainable energy supply. This system encompasses renewable and traditional energy sources, energy storage, and conversion equipment to ensure efficiency and reliability (*Karunarathne et al., 2020*; *Ju, Ding & Hu, 2023*; *Shirvani, 2023*). Illustrated in Fig. 1, the architecture of a comprehensive energy system comprises wind turbines, photovoltaics, combined heat and power (CHP) units, electric and gas boilers, and power-to-gas systems, alongside energy storage systems for electricity, heat, and gas. By harnessing a range of resources such as solar energy, wind energy, biomass, natural gas, and conventional electricity, this system caters to varied energy demands. This diversity mitigates reliance on a singular energy source, thereby bolstering system stability and reliability.

By integrating multi-energy-coupled microgrids into comprehensive energy systems, it becomes feasible to harmonize and optimize the utilization of energy resources, ensuring efficient distribution and utilization of energy within a regional scope. Microgrids operate at the micro-level, comprising small-scale generation and distribution systems that include distributed power sources and energy storage systems. Possessing both power supply and user resources, microgrids can function autonomously and are typically deployed in closed settings such as industrial parks, large facilities, residential communities, or geographically proximate areas. Multi-energy-coupled microgrids, while small in scale, are capable of independent operation and may also connect to the main power grid. This connectivity facilitates the provision of additional power supply to the microgrid when necessary or the transfer of excess energy to the main power grid (*Pan et al., 2022*; *Farah et al., 2021*).

## Microsource optimization configuration in microgrids based on SFLA

In the realm of microgrid optimization, meta-heuristic optimization algorithms like the fireworks algorithm (FWA) and the SFLA are extensively employed to tackle multi-objective and constrained optimization problems. SFLA, a group intelligence algorithm, mimics the process of fireworks explosion to explore the solution space by generating sparks continuously, combining global and local search effectively. Conversely, the frog jumping algorithm emulates the foraging behavior of frog groups, employing global information sharing and local search to find optimal solutions. These algorithms are prevalent in microgrid optimization, enhancing system performance and efficiency by optimizing operation strategies. This study aims to address optimization and operational challenges in multi-energy-coupled microgrid systems by combining the IFWA with the SFLA, thereby enhancing system stability and reliability. IFWA introduces adaptive resource allocation and community inheritance strategies, dynamically adjusting explosion range and spark quantity based on individual optimization states to better meet actual

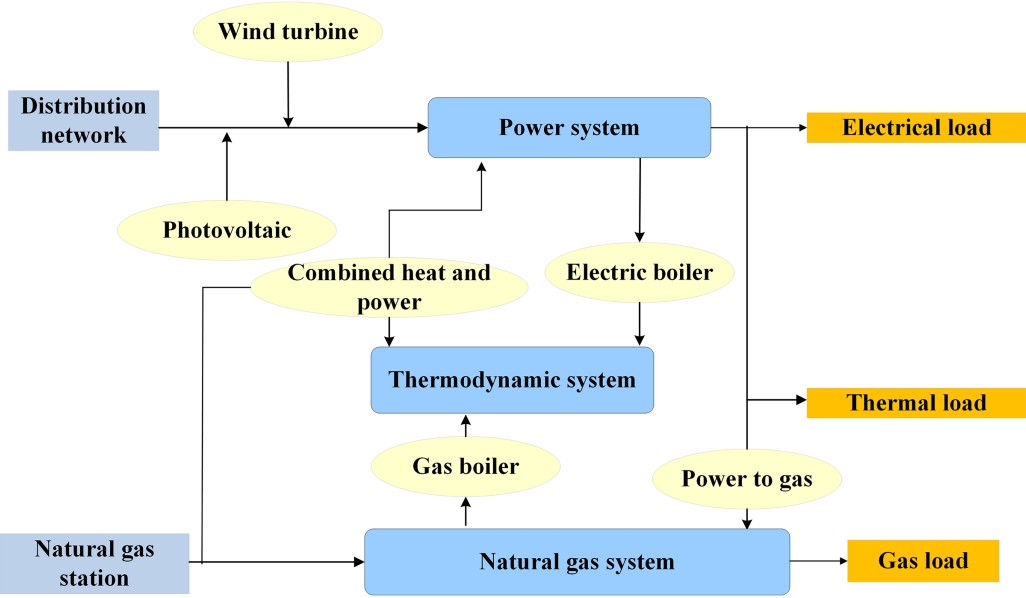

**Figure 1 Architecture of the comprehensive energy system.**

needs. Moreover, utilizing a multi-objective optimization model, incorporating factors like active power network loss and static voltage, the frog jumping algorithm resolves multi-objective optimization problems. Case study simulations conducted in a typical integrated energy system validate the feasibility and superiority of the IFWA-SFLA algorithm.

This study delves into a specific radial distribution network, guided by several assumptions. Firstly, it presupposes a three-phase balanced load that absorbs active and reactive power with specific power factors from the system. Given the network's relatively low voltage level and short distribution lines, mutual inductance between the three-phase lines is disregarded in the analysis. Additionally, the network at the substation is treated as equivalent to a voltage source (*Van Tran et al., 2021*; *Chicco & Mazza, 2020*; *Tran et al., 2021*). Furthermore, the study maintains a constant voltage output model, ensuring that the voltage at the system busbar end remains consistent throughout the analysis.

In microgrid planning, the determination of the location and scale of independent microgrid systems constitutes decision variables for distributed power sources. The optimization objective involves the amalgamation of active power network loss and static voltage. The calculation of active power network loss is depicted in Eqs. (1) and (2):

$$P_{\min} = \min \sum_{k \in N} P_{kloss} \tag{1}$$

$$P_{kloss} = G_{ij}\left(V_i^2 + V_j^2 - 2V_iV_j\cos\theta_{ij}\right) \tag{2}$$

In Eqs. (1) and (2), $P_{\min}$ refers to the active power network loss of the system, $P_{kloss}$ represents the active power network loss of branch k, $V_i$ and $V_j$ denote the voltage at nodes $i$ and $j$, $G_{ij}$ signifies the conductance between nodes $i$ and $j$, and $\theta_{ij}$ denotes the phase difference angle between nodes $i$ and $j$.

The static voltage stability index is computed *via* Eq. (3):

$$V_{stab,k} = \frac{4\left[\left(XP_j - RQ_j\right)^2 + \left(XQ_j - RP_j\right)V_i^2\right]}{V_i^4} \tag{3}$$

In Eq. (3), $P_j$ and $Q_j$ represent the active and reactive power at node $j$, $R$ and $X$ denote the resistance and reactance of branch $k$, and $V_{stab,k}$ signifies the static voltage index of branch $k$.

The weighted factor method integrates the optimization objectives of active power network loss and static voltage, formulated as shown in Eq. (4):

$$f_{min} = \omega_1 \frac{P_{min}}{P_0} + \omega_2 \frac{V_{stab,k}}{V_0} \tag{4}$$

In Eq. (4), $f_{min}$ represents the multi-objective function, $\omega_1$ and $\omega_2$ are the weights assigned to multi-objective allocation, and $P_0$ and $V_0$ respectively denote the initial active power network loss and static voltage of distributed power sources.

The calculation of node voltage constraints is represented by Eq. (5).

$$V_{i,min} \leq V_i \leq V_{i,max} \tag{5}$$

The constraints on distributed power source power are defined by Eqs. (6) and (7):

$$P_{DG\,min} \leq P_{DGi} \leq P_{DG\,max} \tag{6}$$
$$Q_{DG\,min} \leq Q_{DGi} \leq Q_{DG\,max} \tag{7}$$

In Eqs. (6) and (7), $P_{DG\,min}$ and $Q_{DG\,min}$ represent the minimum active and reactive power at the point of distributed power generation connection; $P_{DG\,max}$ and $Q_{DG\,max}$ represent the maximum active and reactive power at the point of power source connection.

The strength of SFLA lies in its amalgamation of global and local searches, enabling the frog population to explore the search space for the global optimum while refining solutions through local search. SFLA comprises two primary components: global information sharing and local search, which partition the virtual frog population into distinct communities. Each community undertakes independent local searches, with individuals gradually improving alongside the community's evolution. As communities progress, frog populations undergo mixing and recombination to form new communities, facilitating the dissemination of local information across the entire population. This iterative process continues until termination conditions are met. This dual-loop mechanism aids in overcoming local optima, bringing the population closer to the global optimum (*Ma et al., 2019*; *Shaheen et al., 2022*).

SFLA employs fitness-based sorting from high to low and utilizes a randomized balanced grouping strategy to enhance the optimization performance of different subgroups (*Xu, Deng & Shen, 2020*; *Vinnikov et al., 2021*; *Osman, Sedhom & Kaddah, 2023*). The operation of this random balancing strategy is as follows: initially, frogs are ranked in descending order of fitness, with individuals possessing the highest fitness

assigned to the first group, followed by frogs with progressively lower fitness levels assigned to subsequent groups. This process continues until frogs with the m-th highest fitness value are allocated to the m-th group. In this approach, based on fitness ranking, (n−1) * m individuals are assigned to the m-th group, while the remaining m frogs are randomly allocated until all frogs are grouped. This strategy effectively moderates the search speed of lower fitness subgroups, thereby enhancing overall search performance and preserving population diversity.

SFLA comprises the following steps (Fig. 2):

1: Initialize parameters by setting the number of subgroups, $m$, the number of frogs in each subgroup, $n$, and determining the total population size, $F$.

2: Create a virtual population by generating $F$ frogs in the space of feasible solutions, $\Omega \subset R^N$, where each frog represents a potential solution, covering the search space for microgrid operational optimization.

3: Sort the frogs using the randomized balanced grouping method to ensure that high-fitness frogs are evenly distributed across different subgroups.

4: Perform optimization operations for each subgroup. All frogs within the subgroups undergo optimization and updates to move closer to the target position. $P_b$ represents the best frog, and $P_w$ represents the worst frog.

5: Conduct local search on $P_b$ to further refine the solutions.

6: Merge all subgroups into the total population $X$ and update the global best frog by aggregating information from each subgroup.

7: Determine whether termination conditions are met. If the conditions are satisfied, output the optimal solution.

## Microgrid energy storage bidirectional inverter control based on IFWA

The circuit structure of the microgrid energy storage bidirectional inverter primarily comprises two modes: grid-connected and islanded. In the grid-connected mode, the bidirectional inverter can link to the large-scale electrical system and operates in charge and discharge modes. Users have the option to select automatic or manual modes. In automatic mode, the battery's charging and discharging process is system-regulated based on preset values. Conversely, manual mode allows users to adjust parameters such as current, voltage, and time for charging and discharging to meet specific requirements. In the islanded mode, the bidirectional inverter can detach from the large-scale electrical system and independently supply power to designated loads. This mode ensures autonomy for local loads, shielding them from external grid disturbances. The bidirectional inverter's performance is influenced by both hardware (such as power electronic devices and energy storage units) and software (including control algorithms and protection strategies). Hence, to maximize the bidirectional inverter's potential, appropriate hardware and software configurations should be selected according to application needs and system limitations.

This study utilizes a voltage-type three-phase bridge inverter, with the main circuit topology depicted in Fig. 3. This inverter serves both rectification and inversion functions. During rectification, energy undergoes rectification and conversion using pulsating

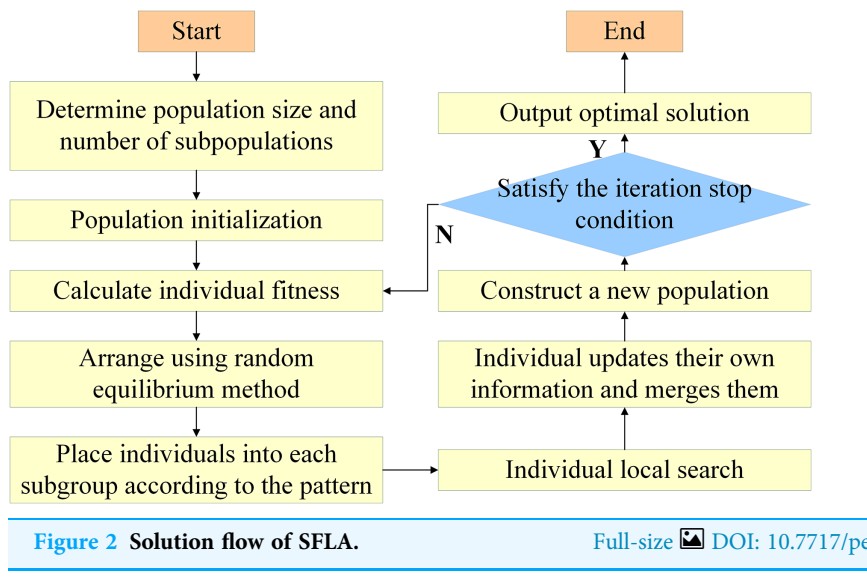

**Figure 2  Solution flow of SFLA.**                         

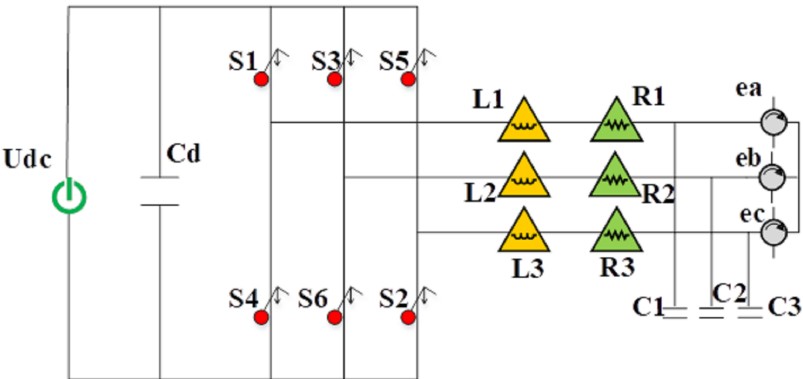

**Figure 3  Topology of energy storage bidirectional inverter main circuit.**

current. Conversely, during inversion, direct current (DC) voltage is inverted and converted using semiconductor switching devices. The operation of this circuit involves modulating the pulse width modulation (PWM) output waveform, facilitating the circuit's transition between inversion and rectification modes for bidirectional energy flow transmission. Power switches S1–S6 and freewheeling diodes D1–D6 constitute the key components. The output side connects to a series resonant filter comprising filter inductance $L_n$ and filter capacitor $C_n$, with $R_n$ representing the inductive impedance. This series resonant filter serves to absorb harmonics introduced by the inverter's switching, preventing these harmonics from affecting power quality within the electrical grid. Additionally, a large parallel capacitor $C_n$ on the DC side is introduced to maintain stable DC voltage and effectively mitigate voltage fluctuations, while also providing a reactive current path.

The droop control characteristics of the energy storage bidirectional inverter resemble the droop characteristics curve of a synchronous generator, demonstrating a proportional correlation between active output and frequency deviation, as well as between reactive

output and voltage deviation. Virtual synchronous generator control mimics the behavior of traditional generators by internally modeling a synchronous generator and adjusting active frequency and reactive voltage to meet control objectives (*Naik, Dash & Bisoi, 2021*; *Tajjour & Chandel, 2023*). Proper selection of control parameters is critical in this process. Correct parameter selection enables the control system to attain desired performance, whereas incorrect choices may degrade system performance or cause failure (*Sharma et al., 2020*; *Iris & Lam, 2019*).

Traditional FWA typically employs fixed parameters to regulate the explosion range and the number of sparks. However, in the improved version proposed in this study, an adaptive resource allocation strategy is introduced. This entails dynamically allocating resources based on the problem's characteristics and the current state of individuals (*Taherian-Fard et al., 2022*; *Singh et al., 2020*). When a community encounters significant optimization challenges, the algorithm can automatically increase resources, such as expanding the explosion range or increasing the number of sparks, to broaden and deepen the search. Conversely, if the community is converging, the algorithm can reduce resources to finely search for the optimal solution. The community inheritance strategy allows individuals to learn and inherit information from their respective communities. This facilitates knowledge sharing, enabling individuals to benefit from each other's discoveries without redundant searches. Consequently, individuals can communicate and collaborate, enhancing the overall efficiency of the algorithm.

In FWA, fireworks with higher fitness generally produce more sparks within a smaller explosion radius, while those with lower fitness generate fewer sparks within a larger radius (*Murugan & Vijayarajan, 2023*). However, allocating resources solely based on the current fireworks' fitness may not always align with the actual optimization scenario. This is because in regions where the objective function's variation trend is steep, fireworks close to the optimal point may exhibit lower fitness, potentially causing smaller sparks within a larger radius to miss the optimum (*Yaprakdal, Baysal & Anvari-Moghaddam, 2019*; *Raut & Mishra, 2023*; *Ramalingam & Shanmugam, 2022*). Conversely, fireworks trapped in local optima often display higher fitness, yet even with multiple sparks within a small radius, they are less likely to escape from local optimal points. In IFWA, an adaptive resource allocation strategy is employed, and the calculation of the $i$-th firework's explosion radius and the number of sparks is as follows:

$$r_i = r/4^{m_i} \tag{8}$$

$$S_i = S + rand(m_i S/M) \tag{9}$$

In Eqs. (8) and (9), $r$ and $S$ denote the initial explosion radius and the number of sparks for fireworks, $m_i$ represents the adaptive coefficient for the $i$-th firework, and $rand(\cdot)$ is a rounding function.

The adaptive resource allocation strategy enables the population to dynamically adjust the explosion radius and the number of sparks based on their evolutionary status. Initially, the population employs a larger explosion radius and fewer sparks to roughly identify

potential optimal locations across a wide range. If a superior location is not discovered after an explosion, the algorithm gradually decreases the radius and increases the number of sparks to conduct a more precise search. This iterative process continues, ensuring that each population converges toward the extremum relatively swiftly.

Another crucial strategy involves promptly eliminating individuals trapped in local optima. This not only prevents resource wastage but also enhances population diversity through suitable mapping strategies. As Gaussian distribution tends to cluster around the original position and exhibits relatively poor exploratory performance, IFWA utilizes the tent chaotic mapping with uniform distribution characteristics to alter the positions of fireworks ensnared in local optima. The fireworks entangled in local optima undergo transformation into the chaotic space across any dimension. The calculation for each dimension $k$ is as depicted in Eq. (10):

$$C_k^1 = (x_k - a)/(b - a) \tag{10}$$

Through multiple iterations using the tent chaotic mapping, the calculation is as shown in Eq. (11):

$$C_k^{l+1} = \begin{cases} 2 \times C_k^1, 0 \leq C_k^1 \leq 0.5 \\ 2 \times \left(1 - C_k^1\right), 0.5 < C_k^1 \leq 1 \end{cases} \tag{11}$$

In Eq. (11), $l$ represents the number of iterations.

The final new position of fireworks is obtained, and its transformation is as described in Eq. (12):

$$x_k = a + C_k^{l+1} \times (b - a) \tag{12}$$

Chaos mapping demonstrates high sensitivity to initial conditions, signifying that even two fireworks confined within the same local optimum area will map to entirely different locations. In contrast, tent chaotic mapping exhibits more uniform distribution characteristics, ensuring coverage of every possible area. The structure of the multi-objective optimization model for multi-energy-coupled microgrids is depicted in Fig. 4. This model encompasses various energy resources such as solar energy, wind energy, battery energy, *etc*., along with loads. Through intelligent control and optimization operations, it achieves synergistic operation and optimal configuration among energy sources. The improved IFWA algorithm, integrated with adaptive resource allocation strategies and community inheritance strategies, dynamically adjusts the range of explosions and the number of sparks based on the optimization status of individuals to meet their actual needs. The construction of the multi-objective optimization model considers active power network losses and static voltage as optimization objectives and utilizes a constrained multi-objective optimization problem-solving approach. SFLA combines global search and local search by optimizing through a virtual frog population. The adoption of a random balanced grouping strategy enhances optimization performance and population diversity.

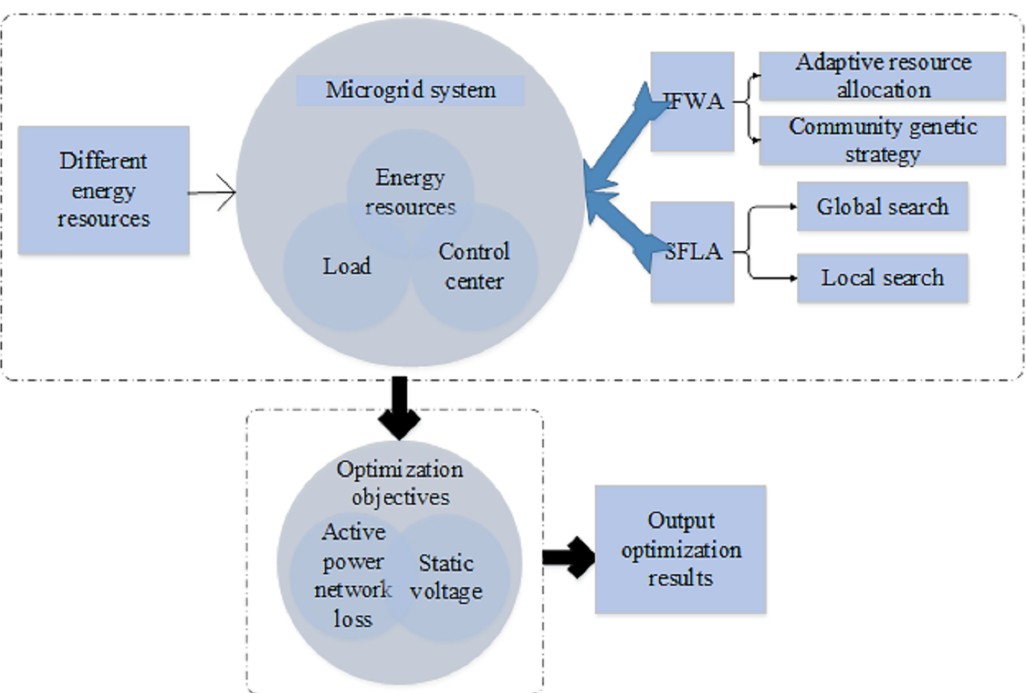

**Figure 4 Structure of the multi-objective optimization model for multi-energy-coupled microgrids.**

The pseudocode for the IFWA-SFLA algorithm is presented in Table 1:

The steps of the IFWA-SFLA algorithm are outlined as follows:

1) Initialization of parameters: The algorithm begins with setting parameters such as the initial explosion range, initial number of sparks, maximum step size, overall scale, number of subgroups, and number of frogs in each subgroup. Additionally, the maximum number of iterations (max_iterations) is determined to regulate the algorithm's execution duration.

2) Iterative optimization process: During each iteration, the algorithm conducts optimization operations on each subgroup. This involves sorting the frogs within each subgroup based on their fitness and distributing them using the random balanced grouping strategy. Subsequently, each frog undergoes optimization to approach the target positions. Following this, local search is performed on the best frog to further refine the solution.

3) Merging and updating: After completing the optimization operations, the algorithm merges all subgroups and updates the global best frog. This step consolidates information from each subgroup into the overall population, ensuring progress towards the global optimum.

4) Termination evaluation: After each iteration, the algorithm evaluates whether the termination condition is met. If so, the optimal solution is outputted, and the algorithm terminates. Otherwise, it proceeds to the next iteration until the maximum number of iterations is reached.

**Table 1 The pseudocode for the IFWA-SFLA algorithm.**

| Algorithm IFWA-SFLA |
| --- |
| Input: Parameters $R$, $M$, $L$, $F$, $m$, and $n$; maximum number of iterations; maximum step size; microgrid main bus voltage; system power loss; static voltage stability index; the number of internal nodes in the microgrid |
| Output: Optimization result |
| 1. Initialize parameters: $R = 300$; $M = 100$; $L = 60$; $F = 150$; $m = 30$; $n = 5$; maximum number of iterations = 50; maximum step size = 488 |
| 2. Create a virtual population: Generate $F$ frogs, with each frog representing a potential solution, covering the search space for microgrid operation optimization. |
| 3. Sort the frogs: Use the random balanced grouping method to sort the $F$ frogs, ensuring that frogs with high fitness are evenly distributed among different subgroups. |
| 4. Perform optimization operations on each subgroup: Optimize and update all frogs within each subgroup to approach the target positions. |
| 5. Perform local search on the best frog to further improve the solution. |
| 6. Merge all subgroups into the overall population $X$ and update the global best frog. |
|   Update the global best solution by integrating information from each subgroup. |
|   - Perform local search on the best frog to further refine the solution. |
| 7. Determine if the termination condition is met. If satisfied, output the optimal solution. |
| End |

## Simulation of IFWA-SFLA

To ensure the reliability and reproducibility of these research findings, meticulous calibration of the hyperparameters of metaheuristic algorithms is conducted, with careful documentation of the utilized values. As highlighted by *Velasco, Guerrero & Hospitaler (2022)*, theoretical limitations within algorithms like genetic algorithms and evolutionary strategies may lead to inaccurate estimations of the global optimal position. The study noted substantial disparities in the landscapes generated by the operators of these metaheuristic algorithms within the solution space, potentially yielding incongruent outcomes when employing extremum theory. To address this concern, the calibration process involved several steps: Firstly, a comprehensive theoretical analysis of the metaheuristic algorithms under consideration is conducted, focusing on the influence of their operators on the solution space and potential theoretical deficiencies. Drawing on this analysis and prior experimental insights, a metaheuristic algorithm well-suited for addressing multi-energy-coupled microgrid optimization problems is identified, and pertinent hyperparameters, such as the crossover and mutation rates for genetic algorithms and the mutation strategy for evolutionary strategies, are delineated. Subsequently, a series of experiments are devised to assess the impact of various hyperparameter combinations on algorithm performance, encompassing diverse problem instances and scales to comprehensively evaluate algorithmic efficacy. Based on the experimental findings, iterative adjustments to the hyperparameters are made, refining their numerical values to enhance algorithmic performance and stability across different settings. Finally, validation experiments using the optimized parameter configurations are conducted to ascertain the algorithm's robustness and efficacy in

**Table 2 Measurement results of algorithm execution time consumption.**

| Algorithm | Start time | End time | Total time (seconds) |
|---|---|---|---|
| PSO | 10:00:00 | 10:05:15 | 315 |
| FWA | 10:00:00 | 10:07:45 | 465 |
| GA-FWA | 10:00:00 | 10:12:30 | 750 |
| IFWA-SFLA | 10:00:00 | 10:04:20 | 260 |

**Table 3 Parameters of the supply units.**

| Equipment type | Power upper limit/kW | Power lower limit/kW |
|---|---|---|
| CHP | 200 | 3,800 |
| Wind turbine | 0 | 1,000 |
| Photovoltaic | 0 | 200 |
| Electric boiler | 0 | 800 |
| Power to gas | 0 | 150 |
| Gas boiler | 0 | 500 |
| Interact with the grid | −1,000 | 4,000 |

solving multi-energy-coupled microgrid optimization problems. Through meticulous analysis of the experimental outcomes, the optimal hyperparameter combinations are identified, and conclusive insights are derived. Aligned with the rigorous approach advocated by *Velasco, Guerrero & Hospitaler (2022)*, the diligent hyperparameter calibration process underscores the reliability and replicability of the research outcomes. In the case study, a typical integrated energy system located in northern China with a scheduling period of T = 24 is considered. The system consists of two CHP units, each with output power limits of 2,000 kW and 100 kW, as detailed in Table 2. Additionally, Table 3 provides the parameters for the energy storage units. In the IFWA algorithm, the parameters are configured as follows: R = 300, M = 100, and L = 60. The population size (F) is set to 150, with 30 subpopulations (m) and five frogs in each subpopulation (n). The maximum number of iterations is capped at 50, while the maximum step size is restricted to 488. Notably, the system's main bus voltage registers at 10 kV, accompanied by active power losses totaling 322.17 kW and a static voltage stability index of 0.113.

To validate the feasibility and efficacy of the IFWA-SFLA algorithm, several alternative algorithms, namely PSO, FWA, and GA-FWA, are employed for comparison. These algorithms undergo testing on the Sphere and Schaffer nonlinear functions, with subsequent analysis of their performance based on experimental outcomes. The Sphere function serves as a classical optimization test function frequently utilized to assess the performance of optimization algorithms. Its mathematical expression is delineated in Eq. (13):

$$f(x) = \sum_{i=1}^{n} x_i^2 \tag{13}$$

**Table 4 Energy storage device unit parameters.**

| Equipment type | Maximum capacity | $P_{c,j}^{max}$ | $P_{f,j}^{max}$ |
|---|---|---|---|
| Energy storage | 500 kW ·h | 80 kW | 80 kW |
| Heat storage | 1,000 kW ·h | 400 kW | 200 kW |
| Gas storage | 375 m3 | 200 m3 | 50 m3 |

In Eq. (13), $x$ represents an n-dimensional vector. Moreover, the Schaffer function is a renowned optimization test function characterized by multiple local minima and one global minimum. Its mathematical expression is portrayed in Eq. (14):

$$f(x) = 0.5 + \frac{\sin^2(\sqrt{x_1^2 + x_2^2} - 0.5}{1 + 0.001(x_1^2 + x_2^2)} \tag{14}$$

In Eq. (14), $x$ denotes a two-dimensional vector.

This study employs the Power Data from the Microgrid dataset (*Bashir et al., 2023*), which encompasses electrical power data collected from the Mesa Del Sol microgrid situated in Albuquerque, New Mexico. The dataset encompasses measurements of voltage, current, power, and energy from microgrid components. It comprises 18 features gathered over the span of 15 months, spanning from May 2022 to July 2023. For experimental training, data from May 2022 to January 2023 is utilized, while data from February 2023 to July 2023 is employed for experimental testing. This dataset holds significance for machine learning applications and holds potential for enhancing microgrid operation and management. The algorithm parameter settings are detailed in Table 4.

## Performance evaluation

To comprehensively evaluate the performance of the IFWA-SFLA algorithm, this study defines and computes the following key metrics. The optimal solution value refers to the numerical value of the best solution found by the algorithm during the iteration process, typically representing the minimum or maximum value of the objective function. Its mathematical expression is shown in Eq. (15):

$$V_{best} = \min_{i=1}^{N} V_i \tag{15}$$

In Eq. (15), $V_i$ is the optimal solution value at the $i$-th iteration, and $N$ is the number of iterations. The average best solution value represents the average of all best solution values obtained after multiple runs of the algorithm, reflecting the stability and consistency of the algorithm across repeated experiments. Its mathematical expression is shown in Eq. (16):

$$V_{avg} = \frac{1}{M} \sum_{j=1}^{M} V_{best,j} \tag{16}$$

In Eq. (16), $V_{best,j}$ represents the best solution value for the $j$-th run, and $M$ signifies the number of runs. The convergence rate indicates the speed at which the algorithm finds

the optimal solution and can be assessed by comparing the number of iterations required for the algorithm to reach the best solution. Its mathematical expression is shown in Eq. (17):

$$CR = \frac{t_{end} - t_{start}}{t_{total}} \tag{17}$$

In Eq. (17), $t_{start}$ is the time when the algorithm starts iterating, $t_{end}$ is the time when the algorithm finds the optimal solution, and $t_{total}$ is the total iteration time of the algorithm. The computation time represents the total time required by the algorithm from start to finish, which is crucial for evaluating the feasibility of the algorithm in practical applications. Its mathematical expression is shown in Eq. (18):

$$T = t_{end} - t_{start} \tag{18}$$

Algorithm stability refers to the consistency of the results generated by the algorithm across multiple runs and can be evaluated by comparing the variation in the best solution values among different runs. Its mathematical expression is shown in Eq. (19):

$$S = \frac{1}{M} \sum_{j=1}^{M} \frac{1}{std_j} \tag{19}$$

In Eq. (19), $std_j$ represents the standard deviation of the best solution values for the $j$-th run, and $M$ is the number of runs. Diversity measures the degree of variation among the solutions in the population, which is crucial for preventing the algorithm from prematurely converging to local optima. Its mathematical expression is shown in Eq. (20):

$$D = \frac{1}{N} \sum_{i=1}^{N} \frac{1}{d_i} \tag{20}$$

In Eq. (20), $d_i$ represents the diversity measure of solutions in the $i$-th iteration, and $N$ is the number of iterations. Standard deviation is the standard deviation of the best solution values, reflecting the range of fluctuation in the algorithm results. Its mathematical expression is shown in Eq. (21):

$$SD = \sqrt{\frac{\sum_{j=1}^{M} \left(V_{best.j} - V_{avg}\right)^2}{M - 1}} \tag{21}$$

In Eq. (21), $V_{best.j}$ represents the best solution value for the $j$-th run, and $V_{avg}$ is the average best solution value. Success rate is the proportion of successful attempts in finding the optimal solution across multiple runs, which is a key metric for assessing algorithm performance. Its mathematical expression is shown in Eq. (22):

$$SR = \frac{C}{M} \times 100\% \tag{22}$$

In Eq. (22), $C$ represents the number of times the algorithm successfully finds the optimal solution.

## EXPERIMENTAL RESULTS ANALYSIS

### Computational analysis of the IFWA-SFLA algorithm

Figures 5 and 6 present the results of various algorithms on the Sphere and Schaffer functions. For the Sphere function (2,500 iterations, 30 dimensions), the IFWA-SFLA algorithm attained an average best solution value of $2.96 \times 10^{-102}$. On the Schaffer function (50 iterations, two dimensions), the IFWA-SFLA algorithm yielded an average best solution value of $3.01 \times 10^{-4}$. Among the compared algorithms, the IFWA-SFLA algorithm demonstrated the highest precision in locating the optimal solution, underscoring its feasibility.

The measurement results of algorithm execution time are presented in Table 5. The data indicates that the IFWA-SFLA algorithm exhibits a relatively shorter total execution time compared to other algorithms, underscoring its efficiency and fast convergence characteristics.

### Content and applications of green building

Figure 7 illustrates the voltage stability index waveform obtained using the IFWA-SFLA algorithm to minimize the system's static voltage stability index as the optimization objective. The results indicate that when minimizing the static voltage stability index is the optimization objective, the optimal connection point is at Node 23. In this case, the active power network loss is 50.093 kW, and the static voltage stability index is 0.014. It is evident that connecting to Node 23 results in the minimum static voltage stability index, demonstrating the successful application of the IFWA-SFLA algorithm in a multi-objective optimal configuration model. Figure 8 presents the amplitude of node voltages. When minimizing the static voltage stability index, the static voltage stability index is minimized, but the active power network loss index is relatively high, possibly not meeting the requirements of the optimization configuration. On the other hand, when minimizing active power network loss is the optimization objective, the active power network loss can be reduced to a minimum, but at this point, the static voltage stability index is relatively high and may not meet the requirements of the optimization configuration. In the multi-objective optimization configuration discussed in this study, individual sub-objectives may not be optimal, but effective adjustments in the relationships between sub-objectives can achieve an overall optimization effect.

The IFWA-SFLA algorithm proposed in this study integrates adaptive resource allocation strategies and community genetic strategies in its design, ensuring efficiency and diversity in the optimization process through a random balanced grouping strategy. These strategies endow the IFWA-SFLA algorithm with powerful global search capabilities and precise local search capabilities, demonstrating unparalleled advantages over other

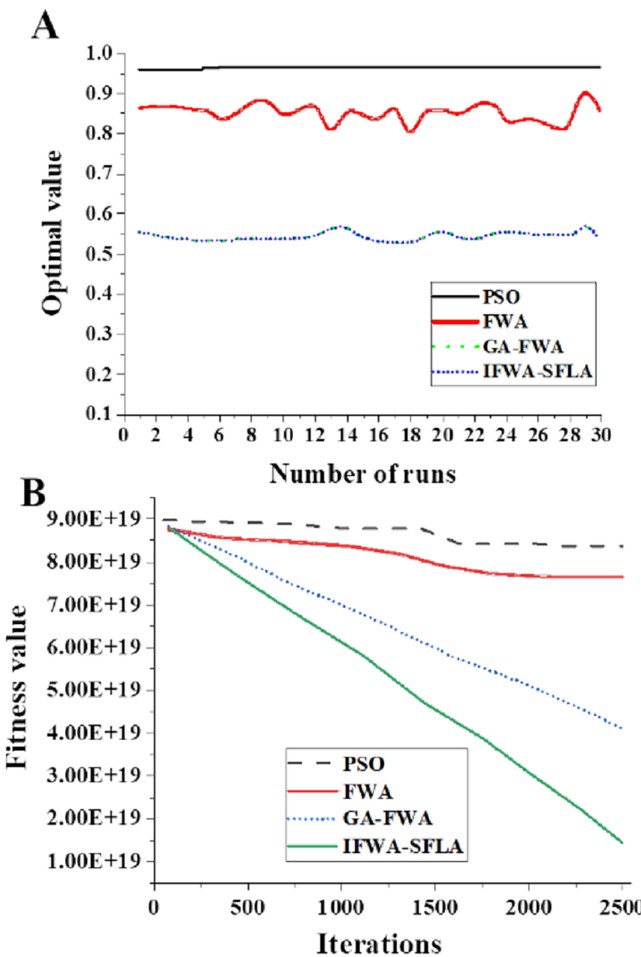

**Figure 5 Running results of different algorithms on the Sphere function.** (A) Optimal value of each operation; (B) global fitness value.

algorithms in the optimization of multi-energy-coupled microgrid operation problems. Simulation experiment results also confirm the applicability and outstanding performance of the IFWA-SFLA algorithm in multi-objective optimization configuration models.

## DISCUSSION

The study employs IFWA in combination with SFLA to address the optimization and operational complexities of multi-energy-coupled microgrids. Unlike many existing metaheuristic algorithms that draw inspiration from natural or artificial processes, the development of this algorithm is specifically tailored to meet the demands of optimizing multi-energy-coupled microgrids. While bio-inspired algorithms, such as those mimicking insect behavior or water flow, have been widely employed in prior research, they may introduce uncertainties and interpretability issues, potentially undermining the scientific rigor of algorithm design. Therefore, metaphorical language is eschewed in favor of a focused approach centered on mathematical modeling and algorithmic design aligned with real-world challenges. The algorithmic framework is informed by a comprehensive understanding and analysis of multi-energy-coupled microgrid systems, coupled with a

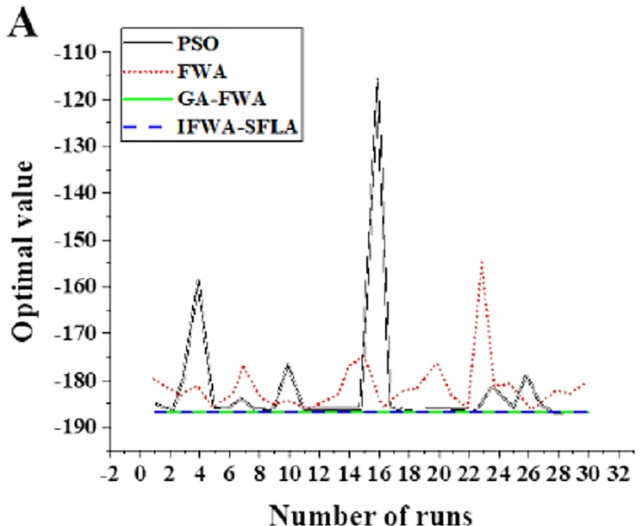

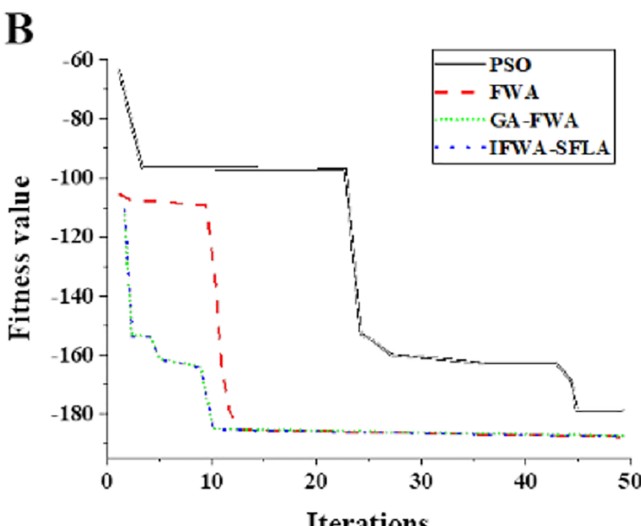

**Figure 6 Running results of different algorithms the Shubert function.** (A) Optimal value of each operation; (B) global fitness value.

**Table 5 Algorithm parameter settings.**

| Algorithm | Parameter | Value |
|---|---|---|
| IFWA-SFLA | $R$ | 300 |
| | $M$ | 100 |
| | $L$ | 60 |
| | Population size | 150 |
| | Number of subgroups | 30 |
| | Frogs per subgroup | 5 |
| | Max iterations | 50 |
| | Max step size | 488 |

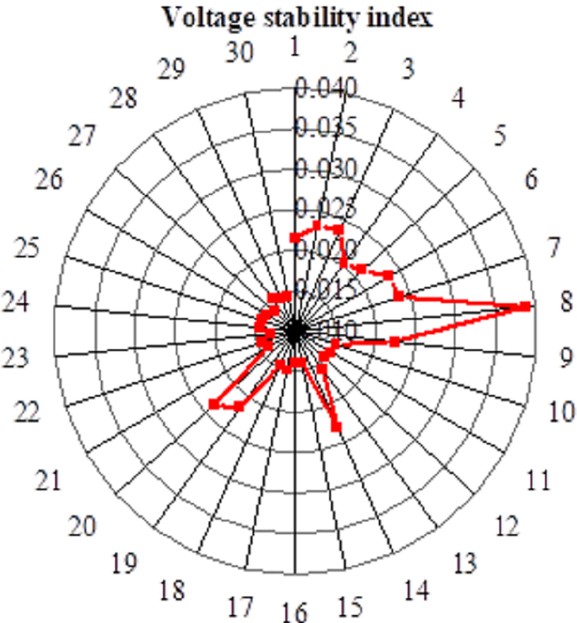

**Figure 7 Static voltage waveform of optimization target.**

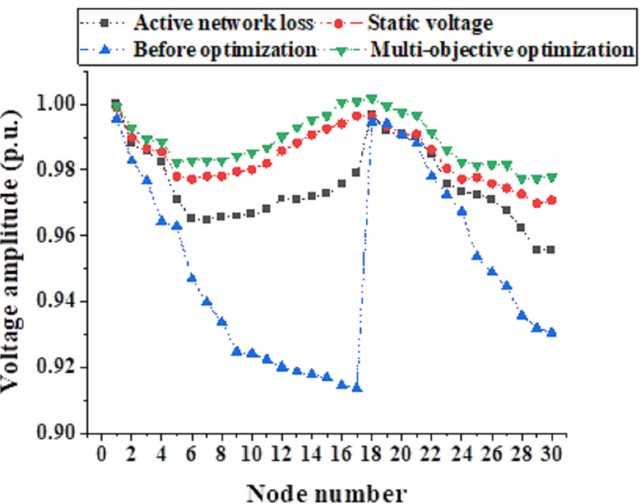

**Figure 8 Node voltage amplitudes.**

meticulous examination of optimization and operational hurdles. Through a systematic evaluation of system requirements and performance metrics, a multi-objective optimization model is formulated, and tailored versions of the fireworks algorithm and shuffled frog-leaping algorithm are devised to address practical challenges. These algorithmic adaptations are meticulously crafted to eschew reliance on metaphors and instead prioritize direct engagement with problem-specific requirements. In adherence to *Sörensen (2015)*'s perspective, emphasis is placed on the exclusion of metaphorical language, opting instead for a scientific and rigorous approach to metaheuristic algorithm design and application. This study underscores the significance of precise problem analysis

and mathematical modeling in crafting solutions for real-world problems, guiding a methodical and interpretable algorithm design process. In previous publications, *Cao et al. (2023)* concentrated on optimizing the planning of multi-energy microgrids (MEMG) during unscheduled islanding events. They introduced a multi-objective stochastic optimization approach to address the economic and reliability concerns of MEMGs during unscheduled islanding events, aiming to resolve outage issues stemming from main grid faults or maintenance. In contrast, the present study is dedicated to tackling the optimization and operational hurdles of multi-energy-coupled microgrids. An IFWA is proposed, integrating adaptive resource allocation and community inheritance strategies to meet the practical demands of microgrid systems. *Jodeiri-Seyedian et al. (2023)* proposed a multi-energy trading microgrid system targeting environmental and economic challenges. Conversely, the current study incorporates considerations of active power network losses and static voltage within a multi-objective optimization model, which is then paired with the IFWA-SFLA algorithm to optimize solutions, enhancing system stability, reliability, and reducing voltage fluctuations. *Sun, Yun & Chen (2021)* developed a robust optimal scheduling model for MEMGs, incorporating uncertainties related to residential biomass waste generation, renewable energy output, and multi-energy loads. Their objective was to minimize operational costs while maximizing waste treatment. In contrast, the current study evaluates the IFWA-SFLA algorithm on nonlinear functions and conducts a case study on a typical integrated energy system, demonstrating the algorithm's feasibility and superiority. In summary, the study effectively addresses the optimization and operational challenges of multi-energy-coupled microgrids by introducing the IFWA-SFLA algorithm coupled with a multi-objective optimization model. This innovative approach enhances microgrid stability, manages electricity flow, and mitigates voltage fluctuations.

The IFWA-SFLA algorithm, introduced in this study, targets the optimization and operational hurdles encountered in multi-energy-coupled microgrid systems, aiming to enhance system stability and reliability. In contrast, *Suresh et al. (2023)* explored various meta-heuristic algorithms, including genetic algorithm, PSO, and mixed integer distributed ant colony optimization (ACO), among others, in their investigation of microgrid system optimization. Their findings favored ACO-based algorithms for their superior convergence time, final solution value, and reliability. However, the IFWA-SFLA algorithm demonstrates enhanced accuracy on nonlinear functions such as the Sphere and Schaffer functions, indicating its efficacy in tackling complex multi-objective optimization problems. Additionally, *Battula, Vuddanti & Salkuti (2023)* delved into the optimization of microgrid energy management systems, utilizing genetic algorithms to address constrained convex problems related to optimal load transfer determination. Their approach yielded reductions in overall cost adaptation by 12.28% and 18.91% in both scenarios, respectively. In contrast, the IFWA-SFLA algorithm successfully identified the optimal connection point at node 23 in a case study by optimizing the minimum static voltage exponent, thereby minimizing the system's static voltage exponent. This underscores the potential applicability of the IFWA-SFLA algorithm in real-world microgrid systems. Furthermore, *Guan et al. (2024)* employed the PSO algorithm to address microgrid optimization

problems, optimizing parameters and control strategies by simulating the collective behavior of birds or fish. While the PSO algorithm demonstrates commendable performance in microgrid optimization, the IFWA-SFLA algorithm offers a more efficient and accurate optimization approach, leveraging the improved spark algorithm and the frog hopping algorithm. Particularly noteworthy is its effectiveness in managing multi-objective optimization problems, where the IFWA-SFLA algorithm adeptly balances the relationships between different objectives to achieve comprehensive optimization. In summary, comparative analysis with the research findings of other scholars highlights the substantial advantages of the IFWA-SFLA algorithm in optimizing and operating multi-energy coupled microgrids. Notably, the algorithm not only showcases high-precision solving capabilities in theoretical test functions but also demonstrates effectiveness and superiority through case studies in actual microgrid systems. Future research avenues may explore further applications of the IFWA-SFLA algorithm in larger and more intricate microgrid systems, as well as its successful integration into real-world microgrid deployments. Additionally, it is imperative to address challenges such as hardware upgrades, costs, and policy and regulatory frameworks that may impact the algorithm's practical implementation.

## CONCLUSIONS

In this study, the IFWA-SFLA algorithm is introduced to address the optimization and operational complexities encountered in multi-energy coupled microgrids, with the overarching goal of enhancing system stability and reliability. The findings underscore the algorithm's notable efficacy in tackling microgrid operational optimization challenges. Firstly, the feasibility and efficacy of the IFWA-SFLA algorithm are corroborated through a case study involving a representative integrated energy system in the northern region. When optimizing for the minimum static voltage index, the algorithm identifies the 23rd node as the optimal connection point, resulting in minimized static voltage index of 0.014. This outcome significantly bolsters the microgrid's operational stability. Secondly, the IFWA-SFLA algorithm demonstrates commendable performance in addressing nonlinear function optimization tasks. On both the Sphere and Schaffer functions, the algorithm achieves impressive average optimal solution values of $2.96 \times 10^{-102}$ and $3.01 \times 10^{-4}$, respectively, indicative of its high precision and robust solving capabilities. Lastly, the study delves into the application of the IFWA-SFLA algorithm within a multi-objective optimal configuration model. Analysis of node voltage stability index waveforms reveals that while individual sub-objectives may not individually optimize, effective adjustment of their relationships facilitates overall optimization. In conclusion, the IFWA-SFLA algorithm emerges as a potent solution for enhancing microgrid stability, managing internal power flow, and mitigating voltage fluctuations. The study outcomes furnish crucial theoretical insights and practical directives for the optimization and operation of multi-energy coupled microgrids. Additionally, the IFWA-SFLA algorithm contributes to the enhancement of microgrid stability by adeptly regulating electrical energy flow within microgrids. This proactive management leads to diminished voltage fluctuations, consequently mitigating the likelihood of power interruptions.

## Limitations

Although the IFWA-SFLA algorithm proposed in this study has demonstrated noteworthy success in optimizing the economic operation of microgrids, it faces certain limitations. Chief among these is the study's primary focus on small-scale microgrid systems, thereby leaving the validation of its applicability to large-scale microgrid systems somewhat incomplete. In future research, several promising directions warrant exploration: Firstly, there is an opportunity for a more in-depth investigation into the performance of optimization algorithms, especially their suitability for addressing the complexities of real-world microgrid systems. Fine-tuning algorithm parameters to enhance their efficacy and robustness in practical scenarios could significantly advance their utility. Secondly, a deeper exploration of control strategies for bidirectional inverters in microgrid energy storage systems, spanning both grid-connected and islanded operation modes, holds promise. Enhancing control algorithms has the potential to bolster the performance and resilience of inverters within microgrids, thereby bolstering the overall reliability and flexibility of such systems. Moreover, exploring the applicability of the IFWA-SFLA algorithm in other domains, such as smart grids and energy management systems, could yield valuable insights. Investigating its efficacy and advantages across diverse contexts may unveil its versatility and benefits beyond microgrid optimization. In summary, future research endeavors should prioritize the refinement and optimization of algorithms and control strategies tailored to multi-energy-coupled microgrid optimization and operation. Such endeavors are poised to drive advancements in microgrid technology, fostering the efficient utilization of energy and furthering sustainable development efforts.

### Funding

This work was supported by the project (Research on the Key Technologies for the Construction and Operation of Multi-Energy Coupled Microgrids Supporting New Power Systems) from Taizhou Hongchuang Electric Group. The funders had no role in study design, data collection and analysis, decision to publish, or preparation of the manuscript.

### Grant Disclosures

The following grant information was disclosed by the authors:
Taizhou Hongchuang Electric Group.

### Competing Interests

Xubo Yue is employed by Taizhou Hongchuang Group. Jing Zhang, Junhui Guo, Jianfei Li, and Diyu Chen are employed by Taizhou Hongyuan Electric Power Design Institute. The authors declare that they have no competing interests.

### Author Contributions

- Xubo Yue conceived and designed the experiments, performed the experiments, performed the computation work, authored or reviewed drafts of the article, and approved the final draft.

- Jing Zhang conceived and designed the experiments, performed the experiments, performed the computation work, authored or reviewed drafts of the article, and approved the final draft.
- Junhui Guo performed the experiments, performed the computation work, authored or reviewed drafts of the article, and approved the final draft.
- Jianfei Li performed the experiments, analyzed the data, prepared figures and/or tables, and approved the final draft.
- Diyu Chen analyzed the data, prepared figures and/or tables, and approved the final draft.

## Data Availability

The code and code description are available in the Supplemental Files.

## Supplemental Information

Supplemental information for this article can be found online at http://dx.doi.org/10.7717/peerj-cs.2139#supplemental-information.

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
