# Peer review of "The optimization and operation of multi-energy-coupled microgrids by the improved fireworks algorithm-shuffled frog-leaping algorithm"

_PeerJ Computer Science, doi:10.7717/peerj-cs.2139_

## Round 0.1 · original submission · Major Revisions

Dear authors,

Thank you for submitting your article. Feedback from the reviewers is now available. It is not recommended that your article be published in its current format. However, we strongly recommend that you address the issues raised by the reviewers, especially those related to readability, experimental design and validity, and resubmit your paper after making the necessary changes. In particular, Reviewer 1's reviews of metaheuristic optimization algorithms are important, and you are expected to address why the new algorithm would be beneficial to this area of research.

Best wishes,

Reviewer 1 ·

Basic reporting

The quality of English is appropriate.
A section on previous works is missing.
The quality of the images is poor and this reviewer considers it unprofessional to provide tables and figures as appendices.
The manuscript is self-contained.
No comment

Experimental design

Study falls within the scope of the journal.
The problem to be solved is clearly defined.
The rigor of the study can be improved.
The methods are described extensively.

Validity of the findings

The novelty of the study is questionable.
No comment
The quality of the conclusions should be improved.

Additional comments

The study proposes an algorithm based on the shuffled frog leaping algorithm (SFLA) and improved fireworks algorithm (IFWA) metaheuristics to solve multi-energy coupled microgrids problems. The study is interesting, although it presents an idea that seems to have been addressed by already published papers. Additionally, multiple aspects that need to be corrected to merit publication were identified. For these reasons, a major revision of the manuscript is recommended.

Comments
Note: In the following comments, several published papers associated with the manuscript's subject matter will be mentioned. It is essential to point out that under no circumstances should the authors consider that the acceptance of their manuscript is conditioned to the citation of the mentioned papers.

It is not necessary for the abstract to provide data on the values of the objective functions achieved by the proposed algorithm, since the resolution of such benchmark problems is not the objective of the study.

There appear to be multiple published studies that address the problem you are considering; this could indicate a lack of innovation in your work with respect to previous studies. Some studies similar to yours that were identified are:
a) Cao, Y., Mu, Y., Jia, H., Yu, X., Hou, K., & Wang, H. (2023). A Multi-Objective Stochastic Optimization Approach for Planning a Multi-Energy Microgrid Considering Unscheduled Islanded Operation. IEEE Transactions on Sustainable Energy.
b) Jodeiri-Seyedian, S. S., Fakour, A., Nourollahi, R., Zare, K., & Mohammadi-Ivatloo, B. (2023). Eco-environmental Impacts of x-to-x energy conversion on interconnected multi-energy microgrids: A multi-objective optimization. Sustainable Cities and Society, 99, 104947.
c) Sun, P., Yun, T., & Chen, Z. (2021). Multi-objective robust optimization of multi-energy microgrid with waste treatment. Renewable Energy, 178, 1198-1210.

Based on the previous point, your study does not mention previous works related to yours. For this reason, it is necessary to include this section in your manuscript.

This reviewer has identified an alarming trend in several manuscripts he has reviewed: many authors include figures and tables as appendices instead of placing them in the body of the manuscript. It has been observed that they do this to complicate the review work or even hide flaws in the results. For these reasons, authors are advised to adhere to higher standards in this subject.
The figures do not have a pleasing aspect ratio, and many are of too low a quality to allow proper interpretation. It is imperative that authors correct this.

Because the study deals with the introduction of a new metaheuristic, the authors should point out the many criticisms against the ever-increasing number of new algorithms, whether they are innovative, improved or combined metaheuristics. The authors can find more information in the following references [1–3].

A serious point of the study: according to Sorensen [4], metaheuristics should be explained without using metaphorical language. The authors are not complying with this rule so necessary to bring academic seriousness to the field of metaheuristics. Please correct.

It has been shown that the hyperparameters of a metaheuristic have a great influence on its performance [5]. Therefore, and in order to improve the reproducibility of their study, it is necessary that the authors indicate the method by which they calibrated the hyperparameters of their metaheuristic and which values were used.

The authors point out that their algorithm has an advantage over others because: "SFLA's advantage lies in its combination of global and local searches, allowing the frog population to traverse the search space to find the global optimum while using local search to refine solutions". However, many algorithms apply this strategy. Therefore, the reasons why better performance could be achieved with the designed algorithm need to be better substantiated.

In the conclusions, the following statement is made: The application of the IFWA-SFLA algorithm significantly enhances the sustainability of microgrids. This contributes to reducing the demand for traditional energy sources, lowering carbon emissions, promoting the use of clean energy, and facilitating the transition of microgrids to sustainable energy sources”. However, these aspects were not studied. Therefore, it is recommended that the authors only provide statements clearly supported by their results.

References
[1] Velasco, L., Guerrero, H., Hospitaler, A. (2023). A Literature Review and Critical Analysis of Metaheuristics Recently Developed, Arch. Comput. Methods Eng., , Doi: 10.1007/s11831-023-09975-0.
[2] Aranha, C., Camacho Villalón, C.L., Campelo, F., Dorigo, M., Ruiz, R., Sevaux, M., Sörensen, K., Stützle, T. (2021). Metaphor-based metaheuristics, a call for action: the elephant in the room, Swarm Intell., (0123456789), Doi: 10.1007/s11721-021-00202-9.
[3] Tzanetos, A., Dounias, G. (2021). Nature inspired optimization algorithms or simply variations of metaheuristics?, Artif. Intell. Rev., 54(3), pp. 1841–62, Doi: 10.1007/s10462-020-09893-8.
[4] Sörensen, K. (2013). Metaheuristics-the metaphor exposed, Int. Trans. Oper. Res., 22, pp. 3–18, Doi: 10.1111/itor.12001.
[5] Velasco, L., Guerrero, H., Hospitaler, A. (2022). Can the global optimum of a combinatorial optimization problem be reliably estimated through extreme value theory?, Swarm Evol. Comput., 75, Doi: 10.1016/j.swevo.2022.101172.

·

Basic reporting

1- Describe dataset features in more details and its total size and size of (train/test).
2- The parameters used for the analysis must be provided in table
3- Add future work in last section (conclusion) (if any)

Experimental design

1- Pseudocode and algorithm steps need to be inserted.
2- Time spent need to be measured in the experimental results.
3- The architecture of the proposed model must be provided

Validity of the findings

1- All metrics need to be calculated n the experimental results.
2- Address the accuracy/improvement percentages in the abstract and in the conclusion sections, as well as the significance of these results.
3- Comparing study results with those of other studies is a crucial step in the research process that authors need to undertake. This is because it facilitates the validation of findings and contributes to the development of a more comprehensive and robust evidence base.

Additional comments

1- Limitation and Discussion Sections need to be inserted.
2- The authors need to make a clear proofread to avoid grammatical mistakes and typo errors.
3- Enhance the clarity of the Figures by improving their resolution.
4- The authors need to add recent articles in related work and update them.

---

## Round 0.2 · accepted · Accept

Dear authors,

Thank you for the revision and for clearly addressing all the reviewers' comments. I confirm that the paper is improved. Your paper is now acceptable for publication in light of this revision.

Best wishes,

Reviewer 1 ·

Basic reporting

The quality of English is appropriate.
The previous work section is appropriate
No comment
The manuscript is self-contained.
No comment

Experimental design

Study falls within the scope of the journal.
The problem to be solved is clearly defined.
The rigor of the study is appropriate
The methods are described extensively.

Validity of the findings

The authors clearly stated the novelty of their study
No comment
The conclusions are adequate and are supported by the results.

Additional comments

This new version demonstrates the authors' commitment to update and improve their work. Likewise, the reviewer's comments were amply and satisfactorily answered. The authors are congratulated for their work since this new version is considered appropriate for publication.

·

Basic reporting

-

Experimental design

-

Validity of the findings

-

Additional comments

Accept.